# Value-Added Products from Catalytic Pyrolysis of Lignocellulosic Biomass and Waste Plastics over Biochar-Based Catalyst: A State-of-the-Art Review

**Peng Li** [1,†]**, Kun Wan** [1,†]**, Huan Chen** [1]**, Fangjuan Zheng** [1]**, Zhuo Zhang** [1]**, Bo Niu** [1]**, Yayun Zhang** [1,*] **and Donghui Long** [1,2,*]

1   State Key Laboratory of Chemical Engineering, East China University of Science and Technology, Shanghai 200237, China
2   Shanghai Key Laboratory of Multiphase Materials Chemical Engineering, East China University of Science and Technology, Shanghai 200237, China
*   Correspondence: yy.zhang@ecust.edu.cn (Y.Z.); longdh@mail.ecust.edu.cn (D.L.)
†   These authors contributed equally to this work.

**Abstract:** As the only renewable carbon resource on Earth, lignocellulosic biomass is abundant in reserves and has the advantages of environmental friendliness, low price, and easy availability. The pyrolysis of lignocellulosic biomass can generate solid biochar with a large specific surface area, well-developed pores, and plentiful surface functional groups. Therefore, it can be considered as a catalyst for upgrading the other two products, syngas and liquid bio-oil, from lignocellulosic biomass pyrolysis, which has the potential to be an alternative to some non-renewable and expensive conventional catalysts. In addition, as another carbon resource, waste plastics can also use biochar-based catalysts for catalytic pyrolysis to solve the problem of accumulation and produce fuels simultaneously. This review systematically introduces the formation mechanism of biochar from lignocellulosic biomass pyrolysis. Subsequently, the activation and modification methods of biochar catalysts, including physical activation, chemical activation, metal modification, and nonmetallic modification, are summarized. Finally, the application of biochar-based catalysts for lignocellulosic biomass and waste plastics pyrolysis is discussed in detail and the catalytic mechanism of biochar-based catalysts is also investigated.

**Keywords:** lignocellulosic biomass; biochar-based catalyst; catalytic pyrolysis; waste plastics; value-added chemicals; fuels

## 1. Introduction

Energy has always been the basis for the survival of humankind, and all human activities are inseparable from the support of energy. However, with the continuous growth of the world population and industrial development level, the human demand for energy has increased dramatically. According to the International Energy Agency estimates, the global energy demand will increase by 35% by 2035 [1]. Therefore, the global energy security issue is facing an increasingly severe challenge. At present, conventional fossil fuels are the main energy sources and essential chemical raw materials [2]. Due to the non-renewable nature of conventional fossil fuels, their massive consumption has left the energy reserves on the Earth dwindling and they will be exhausted at some point in the future. In addition, the use of fossil fuels inevitably leads to the significant emission of greenhouse gases and harmful gases, such as nitrogen oxides and sulfur dioxide [3]. The accumulation of these emissions has led to air pollution and climate change and caused serious global environmental problems that seriously affect human life and the survival of animals. Therefore, the search for alternative fossil fuels and environmentally friendly renewable energy sources can significantly reduce the demand for fossil fuels and thus alleviate the

energy crisis and environmental pollution, thereby reducing the increasing pressure on the Earth. As the only renewable carbon resource, lignocellulosic biomass (LB) is considered a viable energy source for industrial processes [4]. It is a biomaterial, often referred to as a plant or plant-derived material, which is abundantly present in reserves on the Earth and permits the sustainable production of fuels and chemicals with minimal pollution [4,5]. Bioenergy based on LB resources can serve as a substitute for traditional fossil fuels, which can effectively mitigate global energy security issues and significantly reduce greenhouse gas emissions, contributing to alleviating existing ecological problems [6]. Therefore, the use of LB to produce valuable products has attracted the interest of many researchers and has been widely studied.

Catalytic pyrolysis is a technology that enables the efficient utilization of LB resources. Compared with ordinary pyrolysis, it can produce high-quality and value-added chemicals and fuels, namely bio-oil and syngas. Molecular sieve catalysts are commonly applied in LB catalysis due to their unique pore structure and selective catalytic properties, which can catalytically promote aromatic-rich bio-oil production in LB pyrolysis [7]. However, it also faces bottlenecks in use, such as low liquid yields and easy coking, and requires frequent regeneration, leading to increased costs [8]. As the solid product from LB pyrolysis, biochar has a large specific surface area (SSA), well-developed pores, and abundant surface groups, with multiple applications in agriculture, the environment, and energy storage [9]. Moreover, due to the materials' high adsorption capacity, they can also serve as a platform for gas–liquid reactions and are often used as catalysts for many reaction processes, including their application in the catalytic pyrolysis of LB. Biochar can be physically activated, chemically activated, or modified to increase the SSA, the number of groups, and the acidity, which are the physicochemical properties that contribute to the catalytic pyrolysis performance. Moreover, due to the renewable raw materials and low price of biochar-based catalysts (BBCs), they are expected to become a substitute for traditional catalysts. Many researchers have reported the use of BBCs derived from LB for the catalytic pyrolysis of LB to obtain satisfactory chemicals and biofuels [10–14].

In addition to LB, plastic is another type of carbon resource synthesized from petroleum and natural gas, which has the advantages of lightness, high strength, low price, and corrosion resistance, so it has been widely used. However, the casual discarding after use also leads to a large number of waste plastics (WPs) [15]. Traditional disposal methods such as landfill and incineration cannot solve the environmental problems caused by WPs with less pollution and also cause the waste of resources. As with LB, WPs can be converted into $H_2$-rich gas and/or fuel (gasoline, jet fuel, diesel) by catalytic pyrolysis with BBCs to achieve the efficient conversion and utilization of waste carbon resources while solving the problem of WP accumulation. Therefore, this review focuses on the conversion process of LB to biochar and the activation mechanism of activated carbon (AC), the catalytic performance, and role of the obtained BBCs in the catalytic pyrolysis of LB and WPs. Section 2 introduces the formation mechanism, activation, and modification methods of LB-derived BBCs. In Sections 3 and 4, the application of BBCs derived from LB in the pyrolysis of LB and WPs is introduced, respectively. The catalytic mechanism in pyrolysis process is explained. Finally, the problems faced by BBCs and the future outlook are presented.

## 2. BBCs from LB Pyrolysis

### 2.1. LB Composition and Biochar Formation Mechanism

LB contains three main components, known as cellulose, hemicellulose, and lignin [16]. Cellulose is the main component of the plant cell wall and is a natural polymer composed of multiple D-glucose groups. Hemicellulose is a branched-chain polysaccharide around cellulose, composed of several different types of monosaccharides [17]. Lignin is an amorphous polymer closely connected with cellulose and hemicellulose polymers, composed of three phenylpropane units (guaiacol group, purple eugenol group, and p-propylphenol group) connected by C-C and ether bonds [18,19]. In addition to the above three chemical components, LB also includes some inorganic components and extractives [20].

The thermochemical process of LB pyrolysis can generally be divided into four stages. (1) Drying stage. At this stage, the free water of LB feedstocks evaporates at around 100 °C, while the chemical composition remains unchanged. (2) Pre-pyrolysis stage. At 150–275 °C, the chemical composition of LB begins to change. The unstable components, such as hemicellulose, partly decompose into $CO_2$, CO, and a small amount of acetic acid and other substances. (3) Solid decomposition stage. The temperature is in the range of 275–475 °C, which is the main stage of pyrolysis. The LB has a variety of complex chemical reactions, resulting in a large number of decomposition products. (4) Residual carbon decomposition stage. As the temperature rises, the C-O and C-H bonds will be further broken, and the primary pyrolytic oil will also undergo various secondary cracking reactions [21,22]. In fact, it is difficult to delineate the boundaries of the above four stages clearly, and the reactions in each stage may intersect with each other. During these processes, many parallel and tandem chemical reactions occur, such as the decomposition and depolymerization of polymer components, aromatization, condensation, decarbonylation, decarboxylation, dehydration, and demethoxylation [23].

In the pyrolysis process, the three main components in LB have different degrees of thermal stability due to their different compositions and structures. As a result, their pyrolysis behaviors differ. The decomposition of hemicellulose mainly occurs at 250–350 °C, which is represented by xylan. Following this, cellulose decomposition occurs at 325–400 °C [24], and the main pyrolysis product is levoglucosan. Lignin is the most stable component, which is decomposed at 300–550 °C [25]. The decomposition of these three components can produce biochar, but their mechanism is different, so it is necessary to discuss these three components separately. For cellulose, levoglucosan is a considerable intermediate from cellulose pyrolysis [26]; it will form solid biochar through dehydration, decarboxylation, aromatization, and intramolecular condensation. In addition, biochar can be formed by a series of polymerization, aromatization, and intramolecular condensation reactions of levoglucosan [27]. For hemicellulose, furfural is one of the most important pyrolysis intermediates of hemicellulose, which can be converted into solid biochar by dehydration, decarboxylation, aromatization, and intramolecular condensation [20,28]. Cellulose and hemicellulose will form high-aromatic biochar by further decomposition at high temperatures (600 °C) and through an aromatization reaction. The complexity of the lignin structure leads to a more complex decomposition mechanism than in the first two cases. The free radical reaction is one of the principal means and mechanisms of lignin pyrolysis. Lignin can produce many intermediates with aromatic structures, which are essential sources and precursors of biochar [29]. After the biochar is formed, a graphite microcrystalline structure is formed owing to the further cleavage of functional groups and the deep removal of heteroatoms [30].

## 2.2. Activation and Modification of Biochar

To further improve the performance, biochar can be activated or modified [31]. Compared with raw biochar, activated or modified biochar has a larger SSA, more developed porosity, and more active sites or functional groups [32]. In fact, there is no strict boundary between activation and modification, and they both aim to enhance the functionality of the raw biochar. In this review, the activation methods are divided into physical activation and chemical activation. The modification methods of biochar can be divided into metal modification and non-metal modification according to the existence of metal.

### 2.2.1. Activation

#### Physical Activation

The physical activation method is also called the gas activation method, and it is a two-step process [33]. First, the LB is carbonized in an oxygen-free atmosphere at around 400–600 °C to obtain biochar. Then, the obtained biochar is activated by some gases, such as steam, $CO_2$, flue gas, air, and ozone, at an elevated temperature, usually higher than 800 °C. The most commonly used gases are $CO_2$ and steam. The formation of pores in

the physical activation reaction using $CO_2$ as an activation gas can be divided into three stages. The first is to open the pores, which are formed during carbonization but clogged by disordered carbon atoms and heteroatoms. The second stage is the widening, penetration, and deepening of the opened pores. The last stage is the formation of new pores [34,35]. Unlike $CO_2$, steam activation has no pore opening process in the early stages but directly expands the microporous structure of carbon materials, such as by pore expansion [36]. Table 1 lists the pore structure parameters of some ACs prepared from LB by different physical activation methods.

**Table 1.** Pore structure parameters of ACs prepared from LB by different activation methods.

| Raw Materials | Activating Agent | Carbonization Conditions | Activation Conditions | SSA ($m^2$/g) | Pore Volume ($cm^3$/g) | Average Pore Diameter (nm) | Ref. |
|---|---|---|---|---|---|---|---|
| Rice straw | Steam | 500 °C 1 h | 700 °C 1 h | 363 | 0.164 | 1.81 | [37] |
| Barley straw | Steam | 500 °C 1 h | 800 °C 1 h | 534 | 0.299 | 2.24 | [38] |
| Barley straw | $CO_2$ | 500 °C 1 h | 800 °C 1 h | 789 | 0.350 | 1.77 | [38] |
| Coconut shell | $CO_2$ | 600 °C 2 h | 900 °C 4 h | 1700 | 1.135 | 2.65 | [39] |
| Olive-tree wood | Air | 600 °C 2 h | 400 °C 2 h | 481 | 0.272 | - | [40] |
| Hemp stem | KOH | 500 °C 1 h | 800 °C 1 h KOH/char = 4.5 | 2312 | 1.210 | 2.15 | [41] |
| Rice husk | KOH | 400 °C 4 h | 750 °C 1 h KOH/char = 5 | 2121 | 1.022 | 3.71 | [42] |
| Euphorbia rigida | NaOH | - | 700 °C 1 h Impregnation ratio = 1 | 396 | 0.202 | 2.04 | [43] |
| Euphorbia rigida | $K_2CO_3$ | - | 700 °C 1 h Impregnation ratio = 0.75 | 2613 | 1.661 | 2.54 | [43] |
| Euphorbia rigida | $ZnCl_2$ | - | 700 °C 1 h Impregnation ratio = 0.75 | 1115 | 0.640 | 2.29 | [43] |
| Euphorbia rigida | $H_3PO_4$ | - | 700 °C 1 h Impregnation ratio = 1 | 790 | 0.700 | 3.36 | [43] |
| Chestnut | $H_3PO_4$ | - | 450 °C 4 h $H_3PO_4$/chestnut = 0.85 | 584 | 0.357 | 2.49 | [44] |
| Cedar | $H_3PO_4$ | - | 450 °C 4 h $H_3PO_4$/cedar = 0.85 | 374 | 0.251 | 2.52 | [44] |

Specifically, a series of reactions occur during the activation process, resulting in the formation of pores, and the reactions are different when using different activation gases. For example, the main chemical reactions occurring may be the reactions of Equations (1)–(4) when steam is the activating gas. The reaction of Equation (1) mainly occurs below 800 °C, which is the reaction of steam with carbon. Above 800 °C, the reactions of Equations (2)–(4) may occur. Iwazaki et al. [45] found that steam activation increased the SSA of biochar by more than 100 times and also increased the micropore volume. When $CO_2$ is used as the activating gas, the reaction between carbon and $CO_2$ is Equation (2), the required activation temperature is 850–1100 °C, and the activation reaction rate is much slower than that of steam. This is because the diameter of $CO_2$ molecules is larger than that of water molecules; its diffusion speed in the pores of carbon particles is slow, which greatly limits the proximity between $CO_2$ and carbon atoms on the surfaces of micropores. Guo et al. [39] investigated the effect of different condition parameters on the textural characterization of coconut shell AC during $CO_2$ activation. The results indicated that increasing the activation temperature was conducive to the formation and expansion of pores as well as the increase in mesopores. Increasing the activation time is conducive to micropore and mesopore formation, but a longer activation time will lead to the collapse of pores and reduce the

SSA. Increasing the $CO_2$ flow rate is conducive to the reaction of all active sites and pore formation. However, a further increase in flow will cause carbon burnout.

$$C + H_2O \rightarrow H_2 + CO \tag{1}$$

$$C + CO_2 \rightarrow 2CO \tag{2}$$

$$CO + H_2O \rightarrow CO_2 + H_2 \tag{3}$$

$$CO + 3H_2 \rightarrow H_2O + CH_4 \tag{4}$$

In summary, the physical activation process is simple, and there is no chemical solvent involved in the activation process. Therefore, the preparation process is clean, and the liquid phase is less contaminated. The final AC product has a high SSA and a developed pore structure. However, it also has disadvantages, such as a long activation time and high energy consumption, which still represent limitations [46,47].

Chemical Activation

Compared with the physical activation method, the chemical activation method has lower requirements for temperature and time, which is more economical. Furthermore, the AC prepared by the chemical activation method has more pores and can introduce some acid sites and active groups compared with the AC prepared by the physical activation method [48]. Chemical activation can be divided into one-step activation and two-step activation. In the one-step activation process, the raw material and chemical activator are first soaked for a specified time. Then, carbonization of the remaining mixture is carried out after the suspension is dried and activated at a specific temperature to obtain AC. Two-step activation refers to the carbonization of the raw material to biochar, which is then mixed with a chemical reagent, followed by high-temperature activation [49]. Zinc chloride ($ZnCl_2$), potassium hydroxide (KOH), sodium hydroxide (NaOH), phosphoric acid ($H_3PO_4$), and potassium carbonate ($K_2CO_3$) are widely used as activating agents [42,50,51]. In addition, some uncommon activating agents such as $K_2C_2O_4$ and $FeCl_3$ are also used [48]. These activating reagents have one or more of the following effects on the raw material: erosion, hydrolysis, dehydration, and oxidation. KOH, $ZnCl_2$, and $H_3PO_4$ are the most commonly used activation reagents in the chemical activation method. Table 1 lists the pore structure parameters of some ACs prepared from LB by different chemical activation methods.

When KOH, an alkaline activator, is used, the possible activation mechanism is as shown in Equations (5)–(13). During the activation process, the main gas produced is $H_2$, which is caused by the reaction of KOH and char, expressed in Equation (5), and the occurrence of water gas and water gas transfer reactions, represented by Equations (7) and (8). Equations (6) and (9) can be considered as catalytic reactions in the presence of $K_2O$ produced by Equations (5) and (6) [52]. The occurrence of these reactions will lead to the severe corrosion of carbon materials, forming a large number of micropores [53]. The $CO_2$ produced will go through the reaction of Equation (9), most of which will be converted into $K_2CO_3$. At 760 °C, the metal K will precipitate, which is realized in Equations (10)–(12). The production of K will cause more corrosion of carbon materials, which will lead to an increase in the number and size of AC cavities. In addition, at this temperature, a small amount of $K_2CO_3$ will decompose into $CO_2$, as shown in Equation (13). The addition of KOH inhibits tar generation, improves the reaction yield, and accelerates the removal of non-carbon atoms such as N and H. The carbon consumed in the activation process is converted to $K_2CO_3$. After washing, pores appear where the carbon was reacted off, giving the product a large SSA [48,54,55]. Figure 1 shows the KOH activation mechanism of AC.

$$4KOH + C \rightarrow K_2CO_3 + K_2O + 2H_2 \tag{5}$$

$$2KOH \rightarrow K_2O + H_2O \tag{6}$$

$$C + H_2O \rightarrow H_2 + CO \tag{7}$$

$$CO + H_2O \rightarrow H_2 + CO_2 \tag{8}$$

$$K_2O + CO_2 \rightarrow K_2CO_3 \tag{9}$$

$$K_2CO_3 + 2C \rightarrow 2K + 3CO \tag{10}$$

$$K_2O + C \rightarrow 2K + CO \tag{11}$$

$$K_2O + H_2 \rightarrow 2K + H_2O \tag{12}$$

$$K_2CO_3 \rightarrow K_2O + CO_2 \tag{13}$$

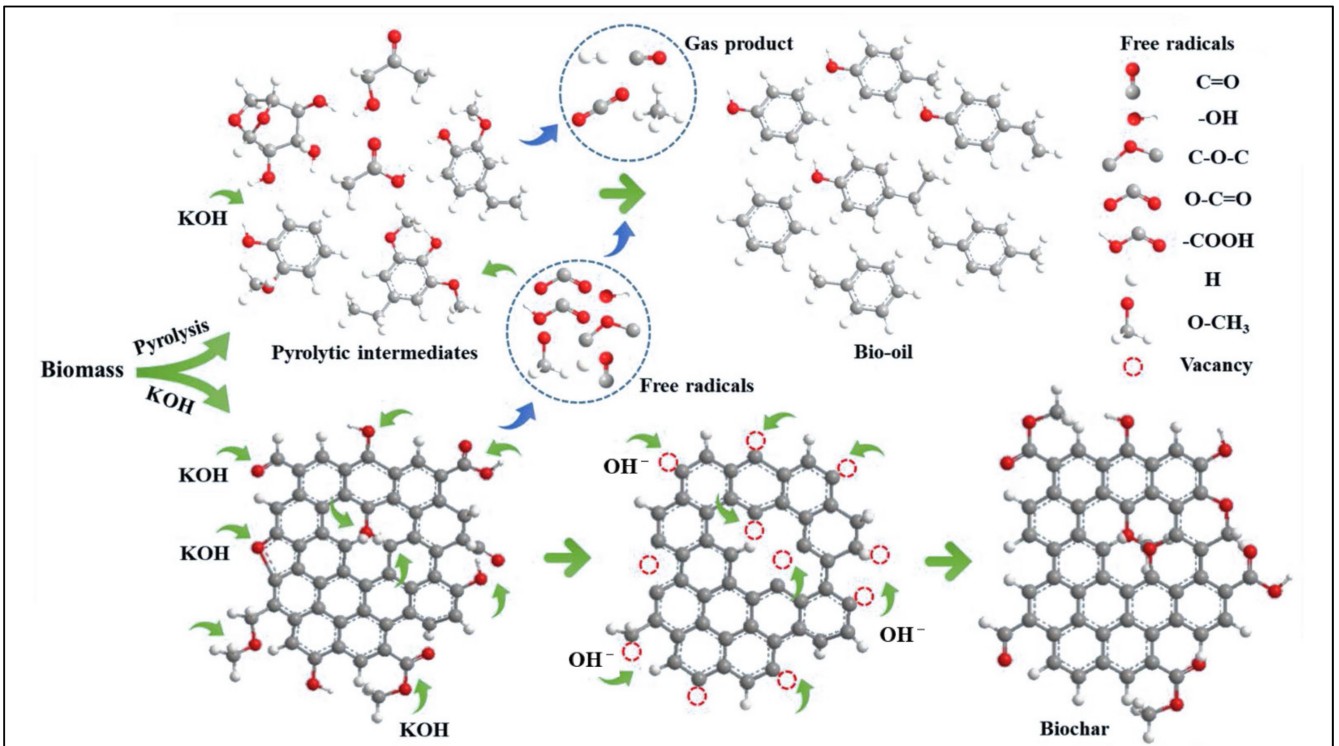

**Figure 1.** KOH activation mechanism of AC from LB [54].

Compared with the KOH activation method, the $ZnCl_2$ activation method has fewer requirements for equipment [56]. In addition, although the SSA of the obtained AC by $ZnCl_2$ activation is lower than that of AC obtained from KOH activation, it can also reach 1500 m$^2$/g, and the required activation temperature is relatively low, usually 500–800 °C. Generally, $ZnCl_2$ is regarded as a type of dehydrating agent in the process of activation [57]. In the process of impregnation, it can dissolve cellulose to form pores. When the concentration of $ZnCl_2$ is high, it will react with water according to Equation (14) to form hydroxy dichlorozinc acid. The resulting acid has an etching effect, which can remove impurities and form a pore structure. During high-temperature carbonization, some $ZnCl_2$ will be converted to ZnO through Equation (15), after which some $ZnCl_2$ or ZnO will remain in the pores, thus changing the pyrolysis behavior and making the tar yield decrease [58]. Because $ZnCl_2$ induces many H and O atoms to be removed as $H_2O$, instead of forming hydrocarbons or oxygen-containing organic compounds, the carbon yield is much higher. In addition, when $ZnCl_2$ and ZnO are removed by acid pickling, the remaining cavities will provide additional porosity. The obtained AC has more micropores because the radius of $Zn^{2+}$ (74 pm) is smaller than that of $Na^+$ (102 pm), $K^+$ (138 pm), and $Ca^{2+}$ (100 pm), which is conducive to the formation of micropores [59]. Figure 2a shows the activation mechanism of $ZnCl_2$ for AC preparation.

$$ZnCl_2 + H_2O \rightarrow H[ZnCl_2(OH)] \tag{14}$$

$$ZnCl_2 + -OH \rightarrow ZnO + HCl \tag{15}$$

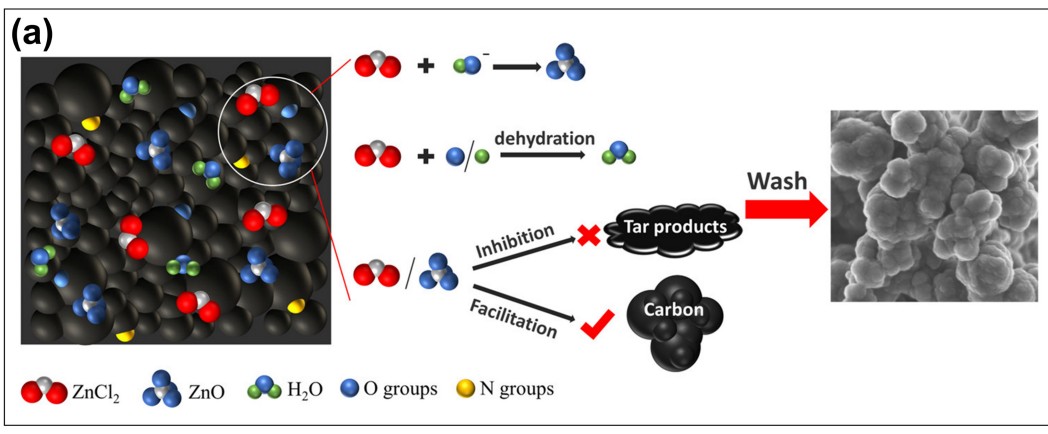

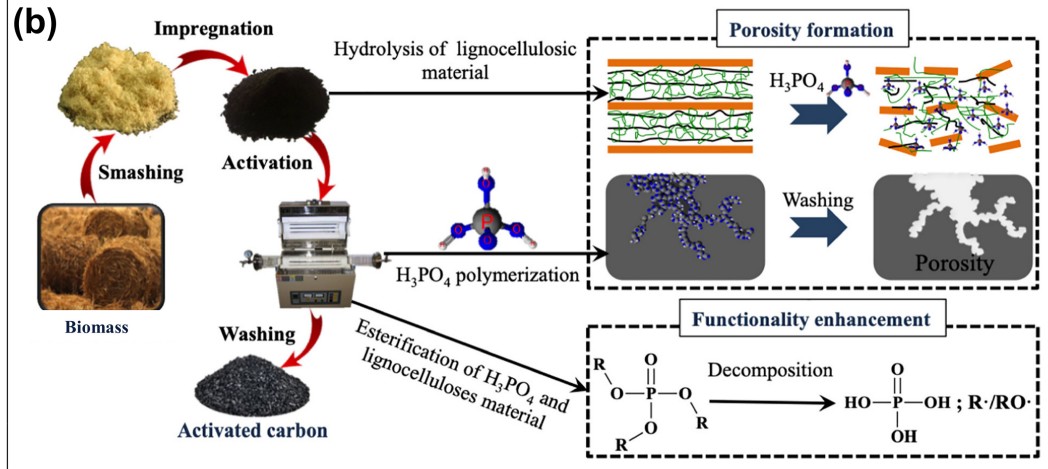

**Figure 2.** Diagram of activation mechanism of (**a**) ZnCl$_2$ [59] and (**b**) H$_3$PO$_4$ for AC preparation [60].

Although the AC prepared by the ZnCl$_2$ activation method is efficient and straightforward, it will cause significant pollution to the environment and affect the health of workers. Moreover, it is also costly, so the activation method is gradually eliminated. Therefore, more researchers are paying attention to the H$_3$PO$_4$ activation method because the activation temperature is generally lower than that of the ZnCl$_2$ activation method and the AC has an abundant mesoporous structure. Furthermore, the prepared AC has strong acidity and many oxygen-containing functional groups (OCFGs) on its surface. In the preparation of AC with H$_3$PO$_4$ activation, there are several mechanisms of action of phosphoric acid on LB. (1) Swelling role. In the impregnation stage, H$_3$PO$_4$ diffuses into LB, resulting in the swelling, hydrolysis, and dissolution of cellulose, hemicellulose, lignin, and polysaccharide into some low-sized molecular sugar monomers, which makes H$_3$PO$_4$ and LB mix evenly [61,62]. Therefore, the raw material can be heated uniformly in the subsequent carbonization process. (2) Dehydration. Some H and O elements in LB are removed as H$_2$O under the action of H$_3$PO$_4$. In this process, water vapor can act as a physical activator to promote the formation of pores [63]. In addition, H$_3$PO$_4$ can inhibit tar formation, thus forming char with high aromaticity. (3) Acceleration of the carbonization activation process. The carbonization process can be accelerated when H$_3$PO$_4$ is used as a medium due to its higher thermal conductivity than some commonly used physical activators [64]. (4) Oxidation. H$_3$PO$_4$ condenses to form different lengths of highly corrosive polyphosphate H$_{n+2}$P$_n$O$_{3n+1}$, leading to more pores and converting micropores into mesopores or macropores [48]. (5) Aromatic condensation. The polycondensation, cyclization, and cross-linking between organic and phosphate ester form a polycondensation

structure at 200–300 °C, which can be transformed into a carbon microcrystal. With the increase in temperature, polyphosphate is converted to $P_2O_5$. As a Lewis acid, $P_2O_5$ can form a C-O=P structure and P-O network, which leads to mesopore formation. Finally, the phosphate or polyphosphate melts and decomposes at 500~600 °C, releasing volatile substances such as $P_4$, $CO_2$, and $H_2O$ [65,66]. When the temperature is higher than 600 °C, the loss of the high-temperature resistance and oxidation resistance of $H_3PO_4$ leads to the erosion of the carbon. In addition, due to thermal shrinkage, $H_3PO_4$ has a tendency to form chemically inert polyphosphate, which remains in the carbon skeleton, thus increasing the ash content of the obtained carbon. (6) Acting as a framework during carbonization. $H_3PO_4$ can act as a framework for carbon generation and deposition. The new carbon and inorganic phosphorus can be bound closely by adsorption bonding. After removing the inorganic components, the inner surface is obtained. Thus, low-dose $H_3PO_4$ produces microporous carbon and high-dose $H_3PO_4$ produces mesoporous carbon. Figure 2b shows the activation mechanism of $H_3PO_4$ for AC preparation.

### 2.2.2. Modification

#### Metal Modification

Metal modification refers to the metal monomers or metal oxides on biochar. The purpose of metal modification is to introduce active sites and improve the catalytic performance. There are two ways to achieve metal modification: (1) mixing metal salts with LB raw materials, and then pyrolysis to produce biochar; (2) pyrolysis of LB raw materials to prepare biochar, and then soaking biochar with metal salts under a specific condition [67,68]. These two methods are widely used in the modification of biochar. The purpose of pre-pyrolysis modification is to improve the SSA of biochar and connect impregnated metal ions to the surface of biochar to provide catalytic active sites for the catalyst [69]. Common metals are iron, nickel, copper, magnesium, and so on. Dai et al. [70] prepared Fe-modified biochar by impregnating rice husk biochar with iron nitrate solution. The modification of Fe can promote micro-mesoporous channel development and improve the catalyst acidity. Cui et al. [71] pointed out that the modification of Fe can also lead to changes in the functional groups of biochar. Zhang et al. [72] mixed rice husk with three metal nitrate solutions and then pyrolyzed it to produce three metal-modified biochar catalysts: Ni–biochar, Fe–biochar, and Cu–biochar. During pyrolysis, carbon atoms in the rice husk biochar would reduce metal ions to produce nickel ($Ni^0$), iron ($Fe^0$), and copper ($Cu^0$) to improve the performance. In general, the modification of biochar by metal impregnation can increase the active sites of the biochar and improve the physicochemical properties.

#### Nonmetallic Modification

Sulfonated biochar is an effective catalyst for acid-driven reactions because it introduces strong sulfonic acid groups, along with the formation of weak acid groups such as carboxyl and phenolic hydroxyl groups [73]. They can catalyze hydrolysis and dehydration processes, as well as esterification and transesterification. Furthermore, $H_2SO_4$ can improve the catalytic efficiency by expanding the SSA and pore structure of biochar. Typical sulfonation reagents include concentrated sulfuric acid (98%), fuming sulfuric acid, and gaseous $SO_3$ [74]. It has been reported that concentrated sulfuric acid sulfonated biochar (99% $H_2SO_4$) exhibits a lower group density of -$SO_3H$ than gaseous $SO_3$ sulfonated carbon, because $SO_3$ gas has more activity and selectivity [75]. In addition, the former carried weak acid as hydrogen bonding sites, which promoted the water adsorption.

In addition to sulfonating biochar, the use of a non-inert gas as a pyrolytic atmosphere during the pyrolysis process can also constitute a nonmetallic modification method of biochar, namely gas purging. Some studies have investigated the modification of biochar with the treatment of $CO_2$, $NH_3$, or their mixtures. Kim et al. [76] used $CO_2$ as the gas medium during oak wood pyrolysis to modify biochar. The SSA and total pore volume of biochar produced in the $CO_2$ atmosphere were twice as high as those in the $N_2$ atmosphere. During pyrolysis, $CO_2$ can react with C in biochar to form CO, thus forming many microp-

ores, which enhance the release of volatile organic compounds during pyrolysis [77]. This method can also be used to dope heteroatoms in the framework of biochar. Chen et al. [78] prepared N-doped biochar by the pyrolysis of bamboo, with a mixture of $N_2$ and $NH_3$ as the gas medium. Compared with the biochar without doping, the SSA and pore volume of the doped biochar increased rapidly. More importantly, the content of N-functional groups on the surface of N-doped biochar increased sharply, while the content of OCFGs on the surface significantly decreased. Table 2 lists the pore structure parameters of modified biochar prepared from LB by different modification methods.

**Table 2.** Pore structure parameters of modified biochar prepared from LB by different modification methods.

| Raw Materials | Pretreatment | Carbonization Conditions | Post-Processing | SSA $(m^2/g)$ | Pore Volume $(cm^3/g)$ | Average Pore Diameter (nm) | Ref. |
|---|---|---|---|---|---|---|---|
| Rice husk | - | 550 °C | 600 °C 1 h 0.4 M $Fe(NO_3)_3$ solution | 110 | 0.050 | - | [70] |
| Rice husk | 0.04 M $Ni(NO_3)_2$ | Microwave power 800 W 20 min | - | 184 | 0.122 | 2.66 | [72] |
| Rice husk | 0.04 M $Fe(NO_3)_3$ | Microwave power 800 W 20 min | - | 193 | 0.103 | 2.15 | [72] |
| Rice husk | 0.04 M $Cu(NO_3)_2$ | Microwave power 800 W 20 min | - | 189 | 0.114 | 2.41 | [72] |
| Pine wood | - | 500 °C 40 min | 600 °C 1.5 h 40% $ZnCl_2$ solution/biochar = 1 | 742 | 0.440 | 2.15 | [79] |
| Pine wood | - | 500 °C 40 min | 600 °C 1.5 h 40% $ZnCl_2$ microemulsion/biochar = 1 | 661 | 0.640 | 3.62 | [79] |
| Pine wood | 0.5 M $Ni(NO_3)_2$ solution | 850 °C 1.5 h | - | 212 | - | | [80] |
| Pine wood | - | 850 °C 1.5 h | 0.5 M $Ni(NO_3)_2$ solution | 25 | 5.810 | - | [80] |
| Rice husk | - | 510 °C 4 s | concentrated sulfuric acid 90 °C 0.5 h | 4 | | 7.70 | [81] |
| Peanuts hulls | - | 400 °C 1 h | 99% $H_2SO_4$ 100 °C 12–18 h | 338 | 0.180 | 1.06 | [74] |
| Peanuts hulls | - | 400 °C 1 h | 99% non-stabilized solid $SO_3$ 6 days | 1 | 0.001 | 0.78 | [74] |
| Oak wood | - | 680 °C 0.1 h 800 mL $min^{-1}$ $CO_2$ | - | 464 | - | 2.00 | [76] |
| Bamboo | - | 600 °C 0.5 h 50 vol% $NH_3$ | - | 255 | 0.271 | 4.26 | [78] |

## 3. Application of BBCs in Catalytic Pyrolysis of LB and WPs

### 3.1. Biochar as Catalyst for Catalytic Pyrolysis of LB

Biochar has a relatively well-developed porous structure, high mineral content (e.g., K, Ca, Cu, and Fe), and abundant surface OCFGs (e.g., -COOH, -OH, and -C=O) [82]. Therefore, it can be used as a catalyst in LB pyrolysis processes to change the composition

and distribution of the products and so improve their quality. In general, the introduction of biochar catalysts will reduce the bio-oil yield but increase the gaseous product yield. This is because biochar promotes the secondary cracking of pyrolytic volatiles, leading to some reactions, such as deoxygenation and bond breaking, resulting in more gaseous products [83].

Many studies on the use of biochar derived from LB as a catalyst for LB pyrolysis have been reported. For example, Dong et al. [84] prepared biochar with bamboo as the LB feedstock and used it for the microwave catalytic pyrolysis of bamboo. The introduction of the bamboo-based biochar catalyst promoted gas production at the expense of the bio-oil yield. The main components of bio-oil are acetic acid and phenol, with phenolic compounds ranging from 20.2 to 26.0%. The phenol content increased from 1.22% without the catalyst to 15.79–24.04% with the catalyst. The addition of biochar reduced the number of compound species in the bio-oil and improved the target product selectivity. In addition, when the biochar loading was 20%, the maximum syngas content was up to 65.13 vol%. Ren et al. [85] prepared corn stove-derived biochar for the microwave pyrolysis of Douglas fir. They found that the introduction of the biochar catalyst also reduced the yield of bio-oil but significantly increased the yield of gaseous products. In addition, the quality of the bio-oil and gaseous product was improved. The bio-oil contained 46 area phenol and 16 area% hydrocarbons. The content of $H_2$ and CO in the resulting gas was 20.43 vol and 43.03 vol%, respectively. Table 3 summarizes the reaction conditions and results of the catalytic pyrolysis of different LB with biochar catalysts.

**Table 3.** Reaction conditions and results of catalytic pyrolysis of different LB with BBCs.

| Catalyst Precursor | Activation/Modification Reagent | Pyrolytic Feedstock | Target Product(s) | Reaction Conditions and Results | Ref. |
|---|---|---|---|---|---|
| Pine sawdust | - | Hardwood | Pyrolysis vapor | Catalytic temperature is 400 °C. The gaseous-phase composition of $H_2$, $CO_2$, and CO is 3.74%, 32.33%, and 23.19%, respectively. | [86] |
| Chlorella vulgaris | - | Chlorella vulgaris | Bio-oil | Using $Fe_{2.5}$/biochar as catalyst, bio-oil yield is 50.10%, energy recovery factor is 64.46. | [87] |
| Corn stover | - | Douglas fir | Bio-oi and syngas | With a catalytic temperature of 480 °C for 10 min, the feedstock to catalyst is 1:1, the concentrations of phenols and hydrocarbons are 46 area% and 16 area%, respectively. The contents of $H_2$ and CO in syngas were up to 20.43 vol% and 43.03 vol%. | [85] |
| Corn stover | - | Douglas fir | Hydrocarbons | The highest amounts of hydrocarbons (52.77% of bio-oil) were achieved at a reaction temperature of 480 °C for 15 min and catalyst/Douglas fir is 80%. | [6] |

**Table 3.** *Cont.*

| Catalyst Precursor | Activation/Modification Reagent | Pyrolytic Feedstock | Target Product(s) | Reaction Conditions and Results | Ref. |
|---|---|---|---|---|---|
| Nanocellulose | - | Douglas fir | Phenolic monomer | At the temperature of 650 °C and biochar to biomass ratio of 3, phenol concentration is 53.77 mg/mL. The concentration of cresols is 44.51 mg/mL, and the volume percentage of hydrogen is as high as 85.32%. | [88] |
| Bamboo | $NH_3$ | Bamboo | Phenols | Catalytic temperature is 600 °C, ratio of biomass to catalyst is 2, pyrolysis time is 30 min, using biochar-N30 as catalyst. The content of phenols is 82%, especially 4-vinyl phenol. with 31% content and 6.65 wt.% yield, as well as 16% 4-ethyl phenol with 3.04 wt.% yield. | [78] |
| Rice husk | $Ni(NO_3)_2 \cdot 6H_2O$ | Rice husk | Syngas | For 50 g rice husk blended with 15 g Ni/biochar catalysts, microwave output power was set to 700 W for 20 min. Gas yield is 53.9%, volume concentration of desired syngas is 69.96%. | [72] |
| Rice husk | $Fe(NO_3)_3 \cdot 9H_2O$ | Corncob | Phenol and cresol | Catalytic temperature is 500 °C and reaction time is 20 min, using $Fe_{0.2}$/biochar as catalyst. The yields and selectivities were 0.2810 mg/g and 60.55% for phenol and 0.0840 mg/g and 32.49% for cresol, respectively. | [70] |
| Coconut shell | Steam | Cellulose, xylan, corncob, and lignin | Furans and phenols | Catalytic temperature is 350 °C for 18 s. The relative peak areas of methylfurans of cellulose, xylan, and corncob are 39.35%, 27.79%, and 26.82%, respectively. The relative peak areas of phenol of lignin and corncob are 53.83% and 12.34%, respectively. | [89] |
| Wood | Steam | Douglas fir | Phenols | The optimized reaction temperature and ratio of catalyst to biomass were 400 °C and 3:1, respectively. Reaction time of 8 min. The phenol selectivity is 74.61%. | [90] |

**Table 3.** *Cont.*

| Catalyst Precursor | Activation/Modification Reagent | Pyrolytic Feedstock | Target Product(s) | Reaction Conditions and Results | Ref. |
|---|---|---|---|---|---|
| Coconut shell | Steam, $Fe(NO_3)_3 \cdot 9H_2O$ | Palm kernel shell | $H_2$-rich bio-gas and phenol | With a catalytic temperature of 500 °C and 2% Fe/AC as catalyst, the concentration of phenol in the liquid is 75.09 area%, while $H_2$ content is 75.12 vol% in the bio-gas. | [91] |
| Coconut shell | Steam, $Cu(NO_3)_2 \cdot 3H_2O$, $SnCl_4 \cdot 5H_2O$, $Al(NO_3)_3 \cdot 9H_2O$ | Xylose, xylan, glucose, cellulose, and pine | Furan compounds | The pyrolysis temperature, catalytic temperature, and reaction time were 500, 300 °C, and 18 s. Using 4Cu–2Al/C as catalyst, the content of furan compounds was 80.6%, and the selectivity of 2-methylfuran, furfural, and furan was 44.0%, 30.0%, and 15.8%, respectively. The 2-methylfuran selectivity of xylose, xylan, glucose, cellulose, and pine was 28.5%, 44.0%, 30.2%, 40.8%, and 56.9%, respectively, and the corresponding furfural selectivity was 52.6%, 30.0%, 35.7%, 23.8%, and 25.1%, respectively. | [92] |
| Corn stover | $H_3PO_4$ | Douglas fir | Phenols | The yield of bio-oil and the concentration of phenols in ex situ upgrading process are 20.03–32.00 wt.% and 4.14–19.76 mg mL$^{-1}$, respectively. The yield of bio-oil and the concentration of phenols in in situ upgrading process are 10.25–25.50 wt.% and 4.14–9.90 mg mL$^{-1}$, respectively. | [93] |
| Sawdust | $H_3PO_4$ | Rice husk | Syngas, phenol-abundant bio-oil | Pyrolysis and catalytic temperature both at 500 °C for 15 min. Phenol content of 65.56% and CO of 56.09 vol%. | [94] |
| Sugarcane bagasse | $H_3PO_4$ | Cellulose | Levoglucosenone | Feedstock to catalyst ratio is 1:5, catalytic temperature is 300 °C. The yield and selectivity of levoglucosenone are 14.7 and 76.3%. | [95] |

| Catalyst Precursor | Activation/Modification Reagent | Pyrolytic Feedstock | Target Product(s) | Reaction Conditions and Results | Ref. |
|---|---|---|---|---|---|
| Sugarcane bagasse | $H_3PO_4$ | Pine wood, poplar wood, bagasse | Levoglucosenone | Under the pyrolysis temperature of 300 °C and AC to biomass ratio of 1:3, the levoglucosenone yields of pine wood, poplar wood, and bagasse are 9.1, 8.3, and 6.2 wt. %, respectively. | [95] |
| Corn stover | $H_3PO_4$ | Douglas fir | Phonels | Phosphoric acid to biomass ratio of 0.8, microwave power of 600 W, and reaction time of 20 min. The yield and selectivity of phenols were 2.46 mg/m and 75%, respectively. | [96] |
| Pine wood | $H_3PO_4$, NaOH | Rice husk | Syngas and tar | Catalytic temperature is 500 °C for 15 min. Gas yield is 0.290 L/g, the content of CO is 66.68%, and the purity of syngas is 71.17%. The relative content of phenols is 74.67%. | [97] |
| Chlorella vulgaris | KOH, $Fe(NO_3)_3 \cdot 9H_2O$ | Chlorella vulgaris | Bio-oil | Catalytic temperature is 650 °C and using 10% Fe/AC as catalyst, the highest HHV and ER values of 31.26 and 71.58 were recorded, respectively. Bio-oil yield is 46.23%. | [87] |
| Bamboo | KOH, $K_2CO_3$, $KHCO_3$, $CH_3COOK$ | Bamboo | Phenols | Catalytic temperature is 600 °C with reaction time of 30 min. Phenol content: biochar–KOH (67%), biochar–$K_2CO_3$ (58%), biochar–$KHCO_3$ (57%) > biochar–$CH_3COOK$ (56%). | [98] |

The formation mechanism of aromatics in bio-oil is shown in Figure 3. There are three main sources of aromatics from LB pyrolysis: cellulose, hemicellulose, and lignin. Cellulose and hemicellulose are first pyrolyzed to form small molecules such as acids, aldehydes, ketones, dehydrated sugars, and furans [99]. These intermediate products are catalyzed by some acidic sites of biochar and undergo oligomerization, decarbonylation, and decarboxylation reactions to form olefins and aromatics. It has been reported that the superheated water present in pyrolysis volatiles can react with OCFGs, resulting in an acidity increase, which then promotes dehydration and decarbonization reactions, etc., enhancing the conversion of pyrolysis intermediates of cellulose or hemicellulose to phenols [97]. The products of lignin pyrolysis are mainly some phenolic compounds and guaiacol. Guaiacol forms phenolic compounds by decomposing O-$CH_3$, and then further decomposes -OH to form aromatics [6,85]. The increase in $H_2$ and CO concentrations in the syngas is mainly due to two reaction mechanisms: $H_2O$ and CO enter the biochar catalyst, and the water gas shift reaction occurs due to the presence of some metals such as Fe and Cu, resulting in an increase in the $H_2$ concentration [100]. Another reason is the dry reforming of methane, where $CH_4$ decomposes or reacts with $CO_2$ to produce large

amounts of $H_2$ and CO. The presence of K in biochar and microwave radiation both favor biogas conversion [85,101]. Therefore, the activity of biochar as a pyrolysis catalyst comes mainly from the inherent acidic sites and some trace metals, which leads to the limitation of its catalytic performance.

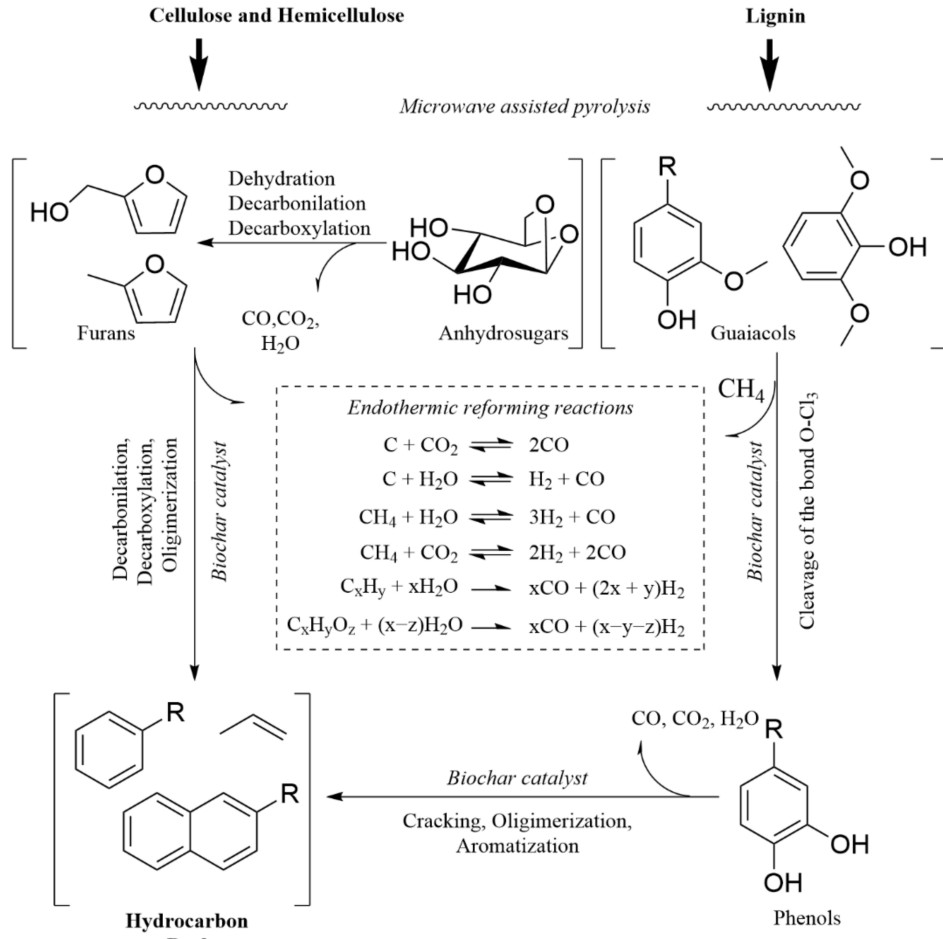

**Figure 3.** Possible pathway of hydrocarbon and syngas production from LB with biochar catalyst [6].

### 3.2. Modified Biochar as Catalyst for Catalytic Pyrolysis of LB

Biochar can be directly used as a catalyst for LB pyrolysis. However, its catalytic activity is limited compared to other pyrolysis catalysts because of its insufficient SSA and limited active centers [102,103]. As mentioned earlier, the presence of more metal elements/ions or functional groups on modified biochar resulted in significantly improved properties. Therefore, many researchers have prepared and used modified biochar as a catalyst for LB pyrolysis.

Dai et al. [70] prepared biochar using rice husk as an LB feedstock and further prepared Fe-modified biochar by the wet impregnation method for the microwave-assisted pyrolysis of corncob. They found that the surface porosity of the biochar was well developed after Fe modification. In addition, the acidic strength of the catalyst surface was increased, facilitating the predicted reaction. The modified biochar reduced the yield of bio-oil and the gas yield was greatly enhanced, which was consistent with the results of unmodified biochar. In the bio-oil, high yields and high selectivity of phenol and cresol were obtained. Meanwhile, the content of ketones and alcohols in the bio-oil decreased significantly, indicating that these oxygenates could be converted into phenols under the action of the catalyst. The content of phenols can reach up to 27.32% when using Fe-modified biochar. Due to the different properties of different metals (electronic properties, redox ability, etc.), different metal-modified biochar materials have different catalytic properties, resulting

in differences in products [31]. For example, Zhang et al. [72] prepared three types of metal-modified rice husk char, Ni–char, Fe–char, and Cu–char, for the microwave catalytic pyrolysis of rice husk in order to obtain the maximum possible gas production and high-quality syngas. Among these three types of metal–char catalysts, the Ni–char catalyst gave the highest gaseous product yield of 53.9 wt.%, with a syngas concentration of 69.96% by volume. The metal content in the metal–char catalyst has a significant impact on the catalytic performance. In conclusion, loading a certain amount of metal on biochar can improve the pore structure of the catalyst and increase the acidity and active centers to facilitate the desired reaction's occurrence in the catalytic process. Table 3 summarizes the reaction conditions and results of the catalytic pyrolysis of different LB with modified biochar catalysts.

The modification of metal biochar is mainly used to increase acidity, and doping heteroatoms on biochar is a good method to introduce non-acidic active sites. After nitrogen doping, the macroscopic and microcosmic properties of the biochar will change significantly, which results in unique electronic conductivity and catalytic performance [104–106]. Therefore, N-doped biochar has attracted the interest of many researchers in pyrolytic LB. For example, Chen et al. [78] prepared N-doped biochar catalysts with different nitrogen content by the pyrolysis of bamboos at different $NH_3$ concentrations, and then used them for the catalytic pyrolysis of bamboo wastes. They found that 4-vinyl phenol is the main phenolic product of bio-oil, and N-doped biochar can promote the production of phenol, inhibit the production of O-containing substances and acetic acid, and release more $CO_2$ and $H_2O$. The N-doped biochar has high catalytic activity as the N-containing groups in the N-doped biochar catalyst are alkaline, which allows them to absorb most of the intermediates produced by pyrolysis. The carbon atom next to pyridine-N, with Lewis basicity, as an active site, can catalyze some intermediate products and promote the formation of phenols. In addition, H atoms on pyrrole-N can also promote the formation of phenols as H donors. The formation of 4-vinyl phenol involves firstly the breaking of the β-O-4 bond of lignin, forming a series of intermediates. Then, these intermediates react with the methoxy group to form 4-vinyl phenol by removing the $CH_2$-OH group. Finally, they further remove the O-$CH_3$ group to form 4-vinyl phenol. In these processes, N-doped biochar provides a large amount of H free radicals to promote these reactions.

### 3.3. AC as a Catalyst for Catalytic Pyrolysis of LB

AC for pyrolysis can be prepared from LB as well as from coal, petroleum coke, and other carbonaceous raw materials. They all have developed pore structures, large SSA, and abundant surface chemical groups [65]. Therefore, AC is widely used in LB pyrolysis. For example, Bu et al. [10] performed microwave pyrolysis of lignin using coal-based AC activated by steam as a catalyst. The main chemical constituents of the resulting bio-oil were phenols, guaiacols, hydrocarbons, and esters. Han et al. [107] also made use of coal-derived AC pyrolysis lignin and found that AC had several different types of acidic OCFGs, which contributed to its surface acidity and effectively promoted single-ring aromatic hydrocarbons' generation.

However, although these ACs have good catalytic activity, compared with the AC derived from LB, the raw materials, such as coal and petroleum coke, are expensive and non-renewable [108]. In the face of increasingly serious energy and environmental problems, researchers are increasingly interested in AC catalysts derived from LB, so it has been extensively studied. For example, Zhang et al. [13] prepared LB-derived AC rich in P-containing functional groups by the phosphoric acid activation method for the selective production of monophenols by the catalytic pyrolysis of cellulose. The resulting AC showed excellent performance, and the selectivity of phenols was up to 99.02%. In addition, they catalyzed the pyrolysis of the monomer of cellulose, glucose, with this AC. The bio-oil was mainly composed of phenols, ketones, and anhydrous sugars, with the highest selectivity of phenols being up to 100%. Figure 4 shows the possible mechanism of phenol formation by the catalytic pyrolysis of glucose and cellulose over AC catalysts. Cellulose pyrolysis

produces large amounts of anhydrosugars, which are subsequently promoted by AC to form some C6 molecules with furan rings, and further rearranged to form a series of phenol intermediates (e.g., 2-cyclopenten-1-one), which are then converted to phenol by the action of AC. On the other hand, the C5 molecules formed by the dehydrated and decarboxylation reactions of the decomposed cellulose and the dehydrated sugars were converted by a series of Diels–Alder, decarbonylation, and oligomerization reactions to the benzene-containing molecules of C6+, which were driven by AC active sites. Some small gaseous molecules are released during this process. The concept of a phenol pool was proposed. Specifically, the -OH of AC, -C-P-O, -P=O, and -P-O functional groups formed a comprehensive system to convert pyrolytic volatiles into phenols instead of aromatic hydrocarbons, and these structures containing benzene (>C6 fraction) were converted into phenol by further decarboxylation, methylation, and bond-cracking reactions. For the remaining two fractions, hemicellulose and lignin, Lu et al. [109] employed coconut shell-derived AC-loaded Pt-based bimetallic catalysts for the catalytic pyrolysis of xylan-based hemicellulose, used to obtain furan compounds. Using 1% Pt-5% Al/AC as a catalyst, the highest content of furans was able to be obtained with 70.51%, mainly including furan, 2-methylfuran, and 2,5-dimethylfuran, with selectivity of 47.95%, 38.49%, and 12.78%, respectively. Duan et al. [11] prepared corn stover-based AC using the phosphoric acid activation method for lignin catalytic pyrolysis and obtained 95.5% selectivity of phenols. It is believed that the mechanism is also mainly the conversion of phenolic compounds to phenols by AC providing a "phenol pool" through decarboxylation, oligomerization, decarboxylation, demethoxylation, and demethylation.

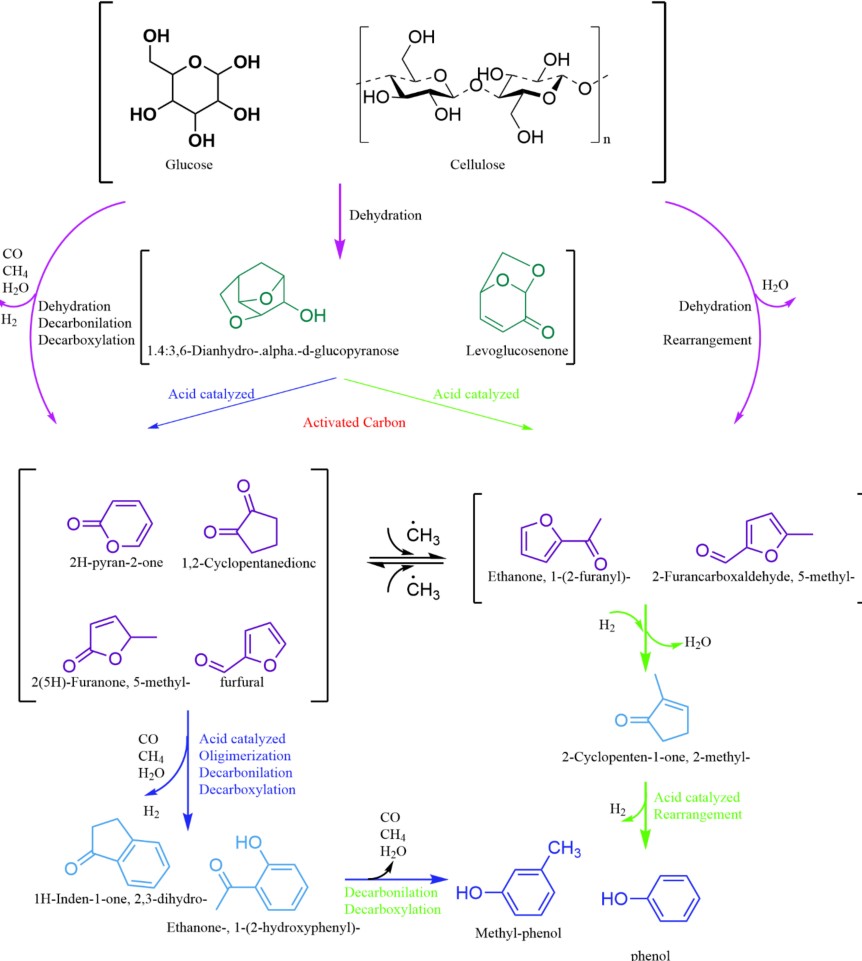

**Figure 4.** The proposed mechanism of forming phenols from catalytic pyrolysis of glucose and cellulose over AC catalyst [110].

Therefore, AC has excellent catalytic performance for the catalytic pyrolysis of individual components of LB, so it also has excellent performance in the catalytic pyrolysis of LB. For example, Ye et al. [95] prepared AC by the $H_3PO_4$ activation method, and catalyzed the pyrolysis of pine wood to selectively produce levoglucosenone. The levoglucosenone yield of pine could reach up to 9.1% wt %, which was much higher than that in the absence of a catalyst (less than 1 wt.%). Yang et al. [96] also prepared LB-derived AC and activated it by $H_3PO_4$. The AC catalyst was used for the catalytic pyrolysis of Douglas fir sawdust to produce alkyl-phenols. The results showed that there were abundant phenolic compounds in the bio-oil because of the good porosity of AC and the catalytic action of the abundant acid surface functional groups. Table 3 summarizes the reaction conditions and results of the catalytic pyrolysis of different LB with AC catalysts.

### 3.4. BBCs for Catalytic Pyrolysis WPs

Municipal solid waste is a major cause of urban environmental problems and can cause significant harm to urban residents and create problems for public health and the environment [111]. WPs have always been a form of municipal solid waste that is challenging to address. The development and innovation of pyrolysis technology provide an excellent way to solve this problem. BBCs still have great potential in this field, and much research has been done. Polyethylene (PE), including low-density polyethylene (LDPE) and high-density polyethylene (HDPE), is a typical waste plastic [112]. Therefore, researchers always use it for pyrolysis to investigate pyrolysis results and mechanisms.

In the absence of a catalyst, the free radical reaction mechanism is the main mechanism of LDPE pyrolysis [113]. The free radical reaction of plastics at high temperatures can be divided into three stages: chain initiation, propagation, and termination [114]. In the chain initiation stage, there are two different reactions. One causes a random break of the chain, resulting in a reduction in the molecular weight of the plastic molecule. The other is the end-break of the chain, where the C-C bond end of the plastic molecule is broken, leading to the production of volatile products. These two reactions result in many free radicals, such as $H\cdot$, $CH_3\cdot$, and $CmHn\cdot$ [115]. In addition, the breakage of the main and side chains occurs, resulting in long- and short-chain radicals, respectively [116,117]. During the propagation phase, these radicals undergo a number of reactions, including hydrogen transfer, β-scission, and isomerization. Hydrogen transfer is the reaction of existing radicals with hydrocarbons to form new radicals and hydrocarbons. β-scission allows the conversion of short-chain radicals to alkanes. Alkanes can also be converted to olefins through dehydrogenation and β-scission. A few alkanes and olefins undergo secondary reactions via hydrogen transfer reactions to form cycloalkanes and cycloolefins, respectively [116,118]. The resulting compounds can further form aromatic hydrocarbons through cyclization and aromatization. In the isomerization stage, the carbon ions tend to form more stable ions by free electron transfer. The C-C bond at this site can be decomposed into olefines and new radicals. In the termination stage, the existing radicals combine to form $H_2$, $CH_4$, short-chain olefins, and alkanes [119].

Table 4 shows the liquid yield and aromatic selectivity from the catalytic pyrolysis of plastics with BBCs. When the catalyst is introduced into the pyrolysis process, the main reaction mechanism is determined by the properties of the catalyst. As an inactive carbon material, the catalytic activity of biochar is mainly determined by the inherent OCFG and pore structure. In addition, the trace metal minerals will also affect its catalytic performance [120]. Figure 5a shows the catalytic mechanism of plastic with biochar. Generally speaking, biochar can promote the cleavage of C-C bonds and generate more short-chain hydrocarbons during the catalytic pyrolysis of LDPE. However, due to the extremely low acidity caused by the limited functional groups on its surface, the reactions promoted by some acidic sites, such as dehydrogenation, hydrogen transfer, and Diels–Alder reactions, occur less, so the yield of aromatics in liquid oil is meager. It has been reported that the presence of plastic during catalytic pyrolysis can promote the secondary pyrolysis of biochar. This is because biochar, as a hydrogen-deficient material, will undergo a hydro-

gen transfer reaction in the process of interaction with hydrogen-rich materials such as polyolefin chains [121]. The dehydrogenation process is enhanced by extracting hydrogen from hydrogen-rich hydrocarbons (such as alkanes) to the carbon surface, resulting in a significant increase in olefin production [122].

**Table 4.** Liquid yield and aromatic selectivity from catalytic pyrolysis of plastics with BBCs.

| Catalyst Precursor | Activating Agent | Pyrolytic Feedstock | Liquid Yield (%) | Aromatic Selectivity (%) | Mono-Aromatic Selectivity (%) | Ref. |
|---|---|---|---|---|---|---|
| Corn stover | / | LDPE | 30.0 | 23.7 | 23.7 | [114] |
| Douglas fir | / | LDPE | 25.0 | 27.7 | 27.7 | [114] |
| Nanocellulose | / | LDPE | 24.0 | 91.4 | 80.8 | [114] |
| Wood chips | / | Mixed plastic | 59.5 | 16.9 | 7.4 | [79] |
| Coconut shell | Steam | LDPE | 64.7 | 19.0 | - | [123] |
| Wood chips | $ZnCl_2$ | Mixed plastic | 43.7 | 42.4 | 2.5 | [79] |
| Wood chips | $ZnCl_2$ | Mixed plastic | 51.8 | 47.6 | 13.7 | [79] |
| Wood chips | KOH | Mixed plastic | 42.6 | 44.7 | 0.54 | [121] |
| Wood chips | $H_3PO_4$ | Mixed plastic | 51.0 | 66.0 | 3.8 | [124] |
| Wood chips | $H_3PO_4$ | PE | 37.5 | 40.0 | 16.2 | [124] |
| Corncob | $H_3PO_4$ | LDPE | 75.3 | - | 54.0 | [125] |
| Corn stover | $H_3PO_4$ | LDPE | - | 29.0 | - | [123] |
| Chestnut shell | $H_3PO_4$ | LDPE | 45.0 | 95.9 | 63.5 | [126] |

AC activated with KOH has a very low mineral content on the carbon surface due to the loss of potassium during the alkali treatment and pre-wash. Therefore, limited hydrogen transfer reactions occur as a result of the lack of metal sites, resulting in a low aromatic content in the oil. KOH activation increases the number of OCFGs on the surface of AC, thus boosting the dehydrogenation step in the process of hydrogen transfer. In addition, the enhanced hydrogen transfer reaction of olefins leads to an increase in the ratio of alkanes to aromatics. The catalytic mechanism of plastic with KOH-AC is shown in Figure 5b.

AC activation by $H_3PO_4$ will introduce numerous acid sites, which will increase the acidity of the catalyst. Zhang et al. [123] used self-made AC with phosphoric acid activation to catalyze the pyrolysis of low-density polyethylene to produce jet-fuel-ranging alkanes and aromatics. The resulting pyrolytic oil contained 48.04 area% alkanes and 28.66 area% aromatics and was within the jet fuel range. Sun et al. [124] prepared phosphoric acid-activated AC for the catalytic pyrolysis of waste PE. Alkanes, olefins, and aromatics are the main chemical components of pyrolytic oil, among which the content of aromatics can reach up to 30.0%. Figure 6a shows the catalytic mechanism of plastic with $H_3PO_4$-AC. The surface of $H_3PO_4$-AC is rich in acidic OCFGs such as -COOH and -OH, and many phosphorus-containing functional groups, such as P=O, $C$-$PO_3$ and $C_2$-$PO_2$. These functional groups can be used as Bronsted acid sites to catalyze hydrogen transfer, cyclization, Diels–Alder, and Friedel–Crafts alkylation reactions [127]. However, the generation of aromatics by the hydrogen transfer reaction will unavoidably produce lots of alkanes, which inhibits the increase in aromatic selectivity. Phosphorus plays an essential role in the catalytic dehydrogenation reaction because it enhances the dehydrogenation activity of other active sites while being able to act as an independent dehydrogenation active site. Therefore, direct dehydrogenation catalyzed by phosphorus-containing functional groups in the catalytic process is an aromatization pathway that consumes both alkanes and olefins, resulting in the reduction of alkenes and alkanes and improving the yield of branched aromatic hydrocarbons.

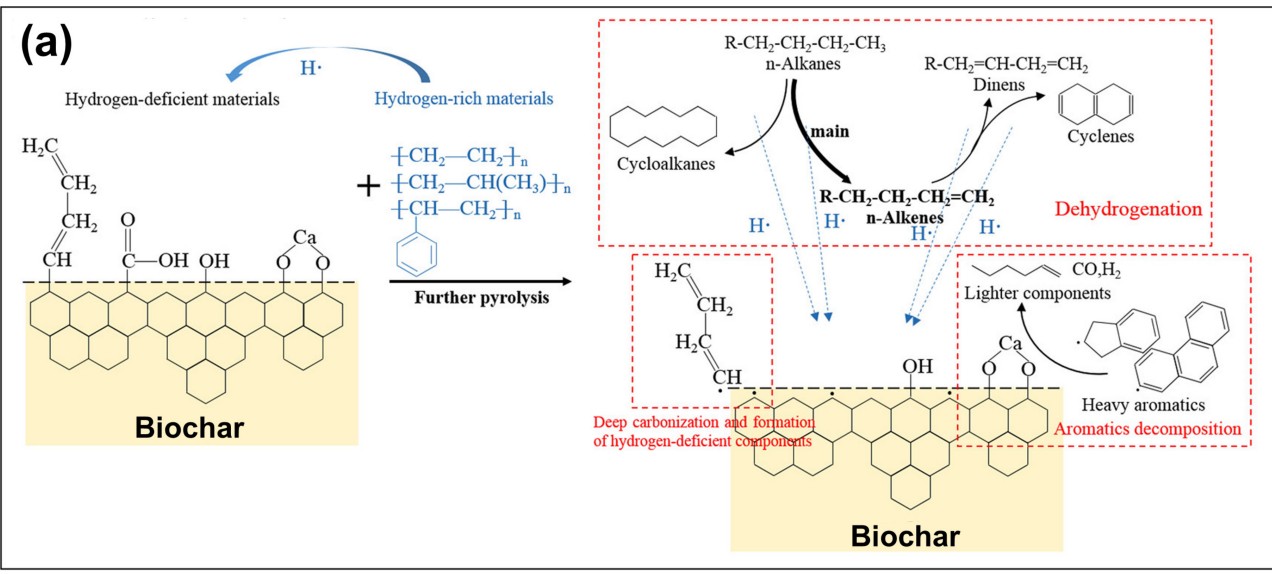

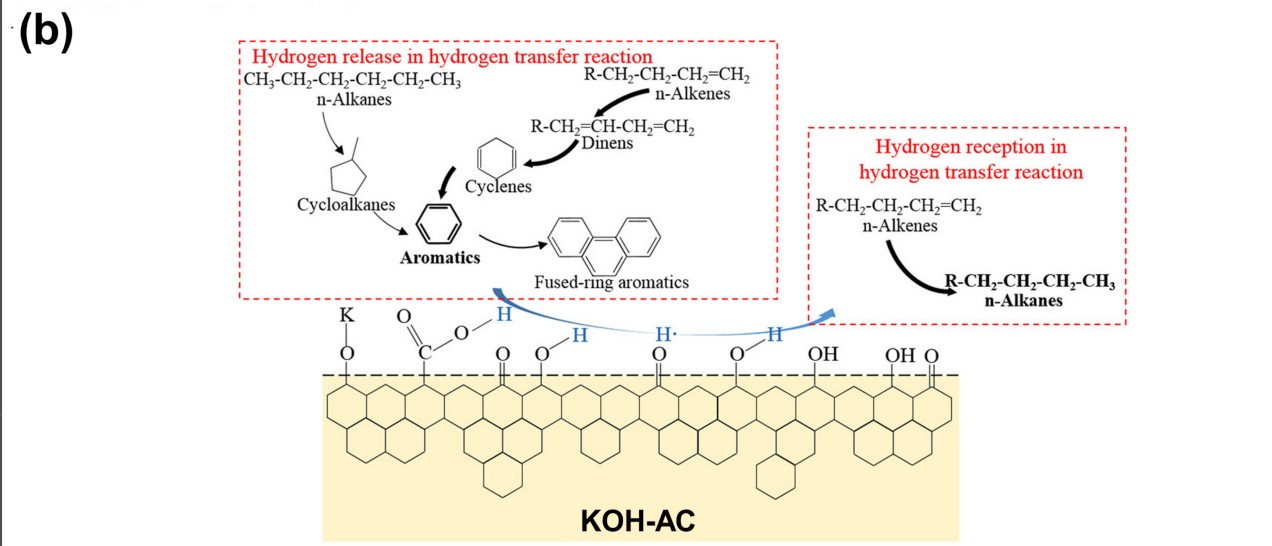

**Figure 5.** Catalytic mechanism of plastic with (**a**) biochar and (**b**) KOH-AC [121].

Similar to $H_3PO_4$ activation, $ZnCl_2$ activation can also increase the acidity of AC, and Zn species can form Lewis acid sites [128]. Sun et al. [79] used AC activated by $ZnCl_2$ for the pyrolysis of WP mixtures (polyethylene, polyethylene, and polystyrene). The content of aromatics in the obtained liquid could reach up to 42.4%, in which bicyclic aromatics accounted for 87.7% of the total aromatics. In addition, the selectivity of 1,3-diphenylpropane in bicyclic aromatic hydrocarbons reached 37%. The presence of Zn facilitated the dehydrogenation and dehydrocyclization of alkanes and olefins, and the aromatization rate was increased through the Diels–Alder reaction, hydrogen transfer reaction, and direct dehydrocyclization. In addition, Lewis acid sites can effectively catalyze the alkylation of aromatics, resulting in the further consumption of olefins and the formation of a large number of aromatics with side chains [129]. In conclusion, $ZnCl_2$ activation leads to the formation of plentiful acid centers (mainly Lewis acid centers), which results in an increase in aromatic content and a decrease in olefin content in the pyrolytic oil. The catalytic mechanism of plastic with $ZnCl_2$-AC is shown in Figure 6b.

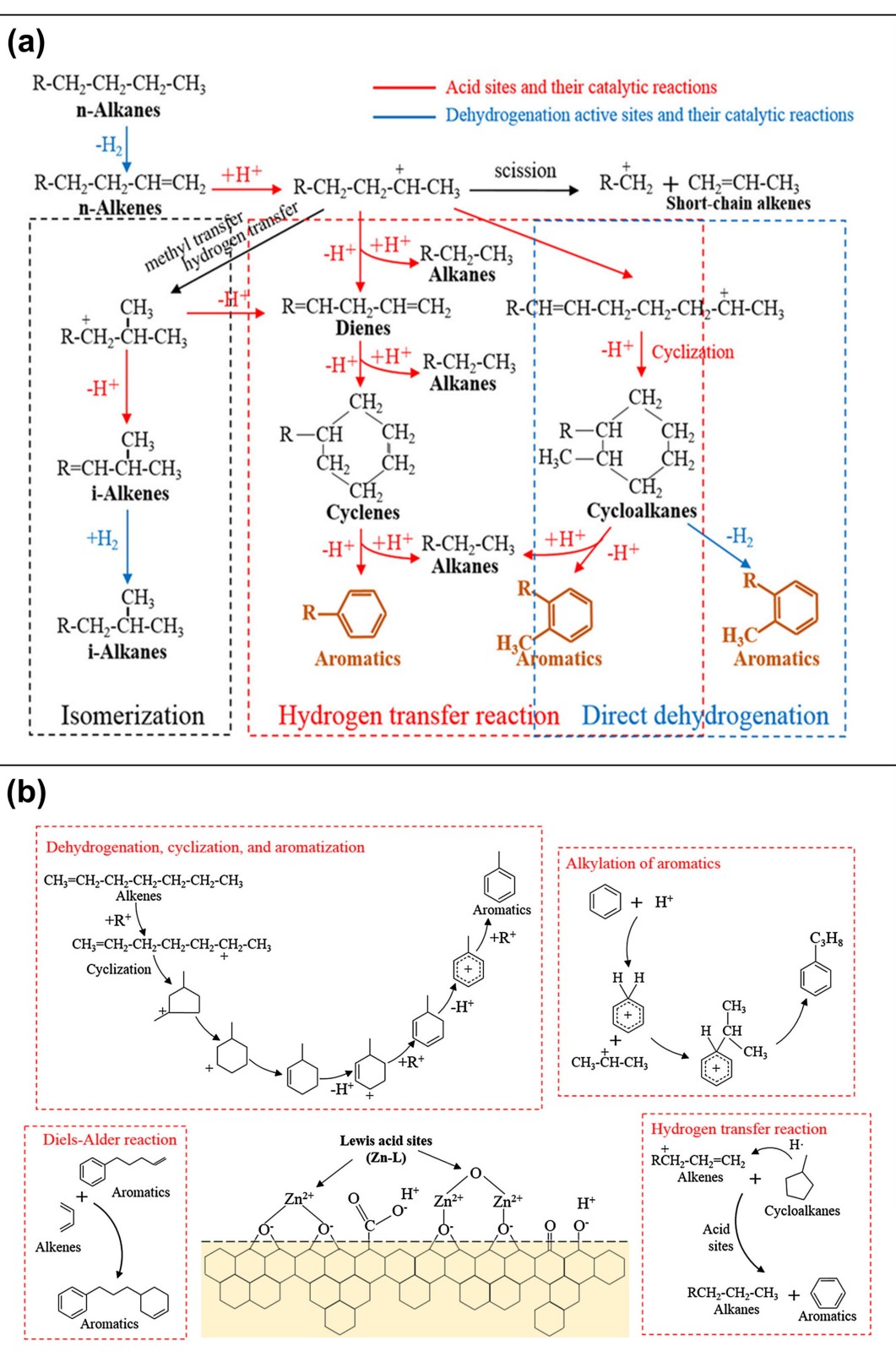

**Figure 6.** Catalytic mechanism of plastic with (**a**) H$_3$PO$_4$-AC [124] and (**b**) ZnCl$_2$-AC [79].

*3.5. BBCs for Co-Pyrolysis of LB and WPs*

In the LB pyrolysis process, one of the purposes of introducing catalysts is to reduce the oxygen content in the bio-oil [130]. From a feedstock perspective, co-pyrolysis of LB

with some substances with high carbon and hydrogen content is a method to increase the calorific value and quality of pyrolytic oil by reducing the oxygen content of the feedstock and creating a synergistic effect. WPs composed of polymers such as polyethylene and polystyrene have high carbon and hydrogen content with low oxygen content and are suitable for co-pyrolysis with LB [131,132]. Studies have indicated that the co-pyrolysis of LB with WPs can increase the yield of pyrolytic oil and is somewhat superior to single-component pyrolysis. Hydrogen-rich plastics provide hydrogen to LB through hydrogen transfer, stabilizing the free radicals generated by LB pyrolysis, thus facilitating the oil conversion of plastics and LB [133]. However, the liquid product remains a complex mixture of oxides and aliphatic hydrocarbons with a wide range of carbon numbers, which is not conducive to further separation and direct use. Therefore, researchers tend to add catalysts to improve the quality of liquid oils, and LB-derived BBCs still have a clear advantage in this field. Table 5 shows the optimal reaction conditions and results of the catalytic co-pyrolysis of LB and WPs with BBCs.

**Table 5.** Optimal reaction conditions and results of catalytic co-pyrolysis of LB and WPs with BBCs.

| Catalyst Precursor | Activating Agent | Biomass | Plastic | Optimal Reaction Conditions and Results | Ref. |
|---|---|---|---|---|---|
| Coconut shell | $H_3PO_4$ | Corn stalk | HDPE | $H_3PO_4$ impregnation ratio = 1:1, carbonization temperature = 700 °C. Aromatics yield = 86.11%, mono-aromatic selectivity = 64.01% | [134] |
| Coconut shell | Steam | Doulas fir | LDPE | Pyrolysis temperature = 500 °C, catalyst/feedstock = 1, LDPE/Douglas fir = 0.7. Jet fuel selectivity= 98.6 area.%, aromatic selectivity= 67.3 area.%. | [135] |
| Corncob | $H_3PO_4$, $H_2SO_4$ | Doulas fir | LDPE | Sulfonation temperature = 100 °C, sulfonation time = 5 h. Bio-jet fuel content= 97.51%. | [75] |
| Corncob | $H_3PO_4$, $Fe(NO_3)_2 \cdot 6H_2O$ | Doulas fir | LDPE | Loading amount = 10%, catalyst/feedstock = 1, pyrolysis temperature = 500 °C. Bio-oil yield = 53.67%, mono-aromatic selectivity = 44%. | [136] |
| Pine sawdust | $Ni(NO_3)_2 \cdot 6H_2O$ | Pine sawdust | LDPE | Loading amount = 10%, residence time = 1.57 min, pyrolysis temperature = 500 °C, catalytic temperature = 600 °C. Hydrogen yield = 392.8 mmol/g | [4] |
| Pine sawdust | $Ni(NO_3)_2 \cdot 6H_2O$ | Rice husk | PE | PE/feedstock = 3. LHV = 14.37MJ/Nm$^3$. | [4] |
| Lignin | $Zn(NO_3)_2 \cdot 6H_2O$ | Lignin | LDPE | Pyrolysis temperature = 450 °C, LDPE/lignin ratio = 12.5%. $H_2$, CO, and $CH_4$ content = 40, 20, 5%. | [137] |

During the catalytic co-pyrolysis of LB and WPs, the LB rich in cellulose and hemicellulose is first pyrolyzed to produce dehydrated sugars (such as l-glucomannan and l-gluconone), followed by dehydration, decarbonylation, and rearrangement to produce a large number of furans. Some furan compounds will be converted into simple phenolic compounds (phenol and methylphenol) and gaseous products (such as $H_2$, $CH_4$, CO, and $CO_2$) via rearrangement reactions [133,138,139]. Some will be combined with plastic primary pyrolysis products such as α-olefins to produce aromatic compounds (mainly monocyclic aromatic hydrocarbons) via the Diels–Alder reaction and dehydration reaction. This reaction path provides a shortcut for preparing aromatic hydrocarbons by catalytic co-pyrolysis and facilitates high-value-added aromatic hydrocarbons. Moreover, the reaction reduces the hydrogen transfer reaction to generate alkanes, releasing olefins and allowing

more olefins to be formed into aromatics through the aromatization reaction. In addition, furan undergoes the Diels–Alder reaction with olefins, which changes the deoxygenation route of furan from decarbonylation and decarboxylation reactions to the dehydration reaction [140]. More carbon in furan is transferred to the aromatic products rather than removed as CO and $CO_2$, which improves the efficiency of carbon atom conversion [141]. Figure 7 illustrates the reaction mechanism of the AC-catalyzed co-pyrolysis of LB and LDPE.

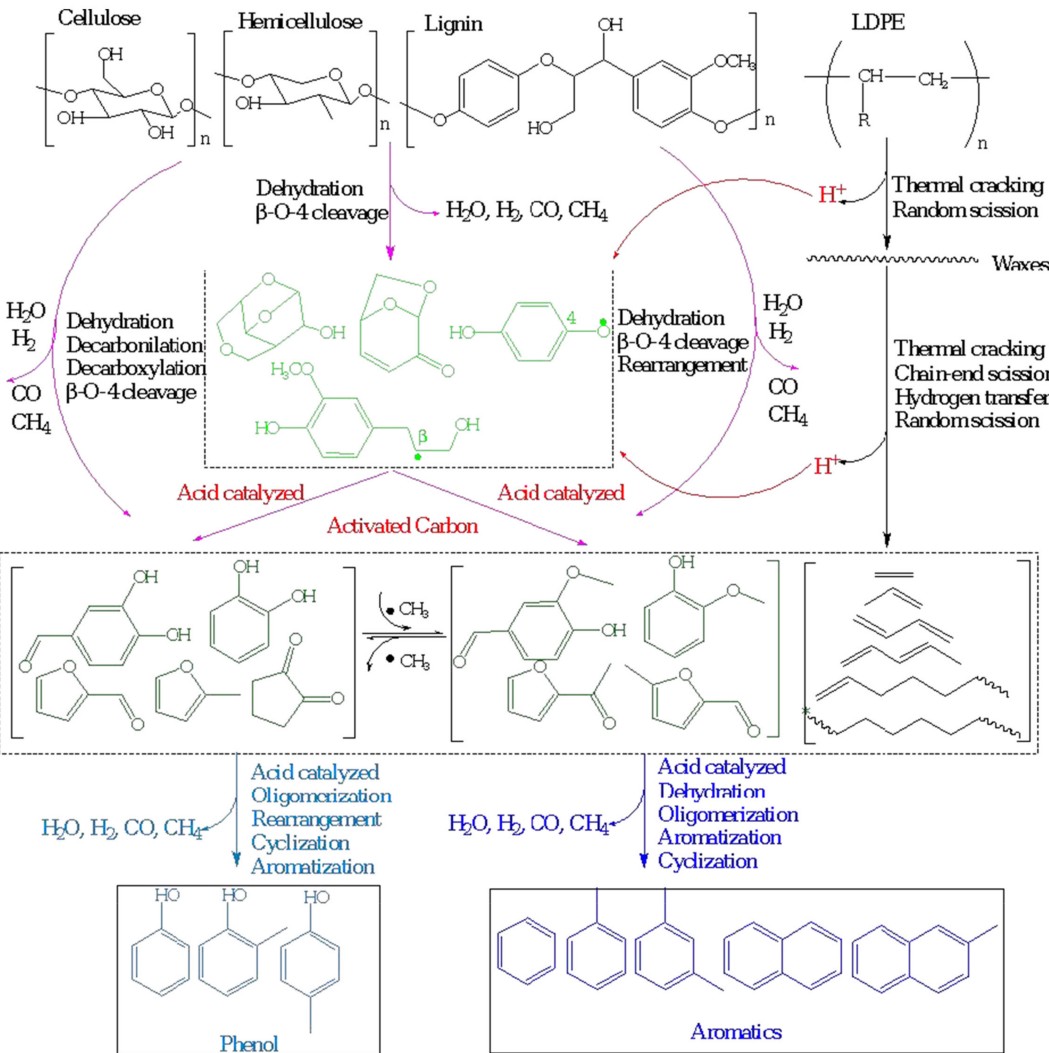

**Figure 7.** Reaction pathway of AC-catalyzed co-pyrolysis of LB and LDPE to phenols and aromatics [135].

When there is no catalyst, the hydrocarbons from plastic pyrolysis can react with the oxygen-containing substances from cellulose pyrolysis. Hydrocarbons provide hydrogen to the oxygen-containing substances, reducing carbon deposition during the deoxygenation process with low hydrogen content. In addition, olefins from plastic pyrolysis may also undergo an oxidation reaction with oxygen-containing substances from cellulose pyrolysis. When these oxygen-containing substances catalyze the rapid pyrolysis alone, they may polymerize independently or with monocyclic aromatics, eventually forming coke deposits [142]. Therefore, the catalytic co-pyrolysis of cellulose and plastic has a significant synergistic effect, which can effectively reduce the formation of coke deposition. Lin et al. [116] prepared an Fe-modified bifunctional AC catalyst for the catalytic co-pyrolysis of Douglas fir and LDPE. The modification of Fe creates new strong acid sites in AC, which promotes the polymerization, cyclization, and aromatization of hydrocarbons. AC promoted the dehydroxylation and demethoxylation of phenols through the hydrogen transfer reaction, and significantly boosted the formation of aromatic hy-

drocarbons. Due to the synergistic effect, the amount of coke is also reduced. Lignin has only a weak synergistic effect with plastic catalytic co-pyrolysis, probably because lignin-derived oxygen-containing substances (such as acids and aldehydes) act as hydrogen receptors and receive part of the hydrogen released from olefins from plastic pyrolysis during aromatization. As a result, some olefins avoid being reduced to alkanes. The main pyrolysis products of lignin are phenolics, which tend to adsorb in the active center to form coke deposition, and cannot react with olefins [143]. In addition, the catalyst can promote the hydrogen transfer reaction and the dehydroxylation of phenol to benzene radicals, which react with olefin radicals to form alkylbenzene. Overall, the yields of high-value-added chemicals from single-component co-pyrolysis with plastics are as follows: cellulose > hemicellulose > lignin.

*3.6. Catalytic Products from BBC Catalytic Pyrolysis of LBs and WPs*

As commented in the previous section, in contrast to the low catalytic activity of conventional biochar, activated or modified biochar-based catalysts can catalyze the pyrolysis of LB and WPs to obtain high-value-added products because of the large SSA and abundant active sites. Therefore, the main research aim of researchers is to obtain high-performance BBC catalysts by increasing the active sites, increasing the SSA of catalysts, or improving the pore structure of catalysts through activation and modification, which can catalyze the pyrolysis of LBs and WPs to obtain phenols, high quality bio-oil, syngas value-added products, and so on. Zhang et al. [123] firstly proposed that the catalytic pyrolysis of plastics on activated carbon activated by phosphoric acid could yield jet-fuel-like pyrolysis oil; then, Wan et al. [125] investigated the catalytic performance of LDPE catalyzed by corn cob-activated carbon activated by different phosphoric acid concentrations, and the results pointed out that phosphoric acid and corn cob at optimal ratio conditions could improve the yield of hydrocarbon products in the jet fuel range in pyrolysis oil. Li et al. [144] investigated N-doped activated carbon (NAC, prepared from walnut shell) for the catalytic pyrolysis of sugarcane bagasse, where the main product was 4-ethyl phenol (4-EP). Approximately 3.03 wt% of 4-EP was obtained on NAC, while the 4-EP yield on activated carbon was only 2.46 wt%, indicating an increase of 23.17% by N-doping modification. Yang et al. [145] used a sulfonated carbon catalyst to catalyze the pyrolysis of cellulose to obtain levoglucosenone (LGO), pointing out that density flooding theory calculations showed that the sulfonic acid group could significantly promote the 1,2 dehydration reaction in glucose pyrolysis, and the selectivity of LGO was increased from 0.5 to 46.4%. Gupta et al. [146] studied the effect of nickel-activated biochar on the pyrolysis of pine needle biomass and concluded that the activation energy (Ea) had been reduced from 25.95 kJ/mol in noncatalytic pyrolysis to 20.79, 15.20, 10.52, 13.99, and 9.69 kJ/mol for BC, Ni/BC, Ni/BC-$ZnCl_2$, Ni/BC-$H_3PO_4$, and Ni/BC-NaOH catalytic pyrolysis, respectively.

## 4. Conclusions and Prospects

This paper reviews the mechanism of the thermochemical synthesis of BBCs from LB, the effects of different activation and modification conditions on their performance, and their application and action mechanisms in LB and plastics pyrolysis. The three major components of LB have different biochar formation mechanisms during pyrolysis, among which lignin has the highest biochar yield. Biochar can be physically and chemically activated to improve its physicochemical properties, such as creating a large SSA and well-developed pores or increasing the surface functional groups for enhanced surface acidity. In addition, other modifications, such as the loading of metal active centers and doping with heteroatoms, can be used to improve its catalytic properties and give it additional functionality. It is necessary to consider the significant differences in the physicochemical properties of ACs obtained from different activator treatments, and the economics and environmental impacts of the activation and modification processes. BBCs have shown good catalytic performance in LB and plastics pyrolysis and the co-pyrolysis of LB with plastics to produce valuable chemicals or biofuels. Due to the renewable

feedstock, low cost, and simple synthesis process, they show good catalytic performance in the field of LB catalysis and are a potential alternative to traditional non-renewable and expensive catalysts.

The catalysts are obtained from biomass and used for the resource recovery of biomass and waste plastics without additional carbon emissions, in line with the current environmental policies of the international community. This can reduce the demand for non-renewable resources while solving the problem of waste pollution. The catalysts can be applied to a wide range of feedstocks, and the pyrolysis of different feedstocks can yield different target products, including liquid bio-oils, aromatic compounds (benzene, toluene, phenol), syngas, and carbon materials. This can increase the production capacity of specific chemical raw materials, which is important for the optimization and upgrading of the product structure and industrial structure. The low cost, simple process equipment, easy availability of raw materials without relying on imports, and high market demand for the products mean that pyrolysis value-added products have good market prospects.

Despite the excellent catalytic performance of BBCs, their role in many reactions has not been well studied and elucidated. Future research should be devoted to explaining the detailed catalytic mechanism of BBCs and identifying the relationship between their physicochemical properties and catalytic performance. Since the physicochemical properties are closely related to the preparation and modification conditions, some still unknown activation or modification mechanisms need to be investigated in depth to determine the optimal preparation conditions and modification methods for different reactions, and it is necessary to evaluate multiple modification methods to improve the catalytic activity and reduce unnecessary intermediate processes and the wastage of reagents. In addition, most of the current studies on BBCs have been conducted in the laboratory and they have not been tested in more environmentally complex chemical processes. For application in practical industrial applications, it is necessary to consider additional factors that can affect the activity and stability of the catalyst during preparation and catalysis. The deactivation, regeneration, recycling, and reprocessing of BBCs should also be investigated in order for them to be fully utilized and to reduce their environmental effects.

**Author Contributions:** Conceptualization, P.L., K.W., Z.Z. and Y.Z.; methodology, P.L. and K.W.; software, P.L., K.W. and H.C.; validation, P.L., K.W. and Z.Z.; formal analysis, Y.Z.; investigation, H.C., B.N. and F.Z.; resources, Y.Z.; data curation, P.L.; writing—original draft preparation, P.L.; writing—review and editing, P.L.; visualization, P.L. and K.W.; supervision, Y.Z.; funding acquisition, Y.Z. and D.L. All authors have read and agreed to the published version of the manuscript.

**Funding:** This research was funded by the National Natural Science Foundation of China (No. 22008073) and the Shanghai Sailing Program (20YF1410600).

**Data Availability Statement:** Not applicable.

**Acknowledgments:** Reproduced from Refs. [6,110,135] with permission from the Royal Society of Chemistry. Reprinted (adapted) with permission from Refs. [121,124], Copyright © 2018, American Chemical Society.

**Conflicts of Interest:** The authors declare no conflict of interest.

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
