# Peer review of "Value-Added Products from Catalytic Pyrolysis of Lignocellulosic Biomass and Waste Plastics over Biochar-Based Catalyst: A State-of-the-Art Review"

_catalysts, doi:10.3390/catal12091067_

Round 1

Reviewer 1 Report

This review is well written and relatively complete in all aspects. It is recommended to receive it after minor repairs.

I only found one place that needs to be modified, as follows:

line 291, "The modification of Fe can promote micro-mesoporous channels development, and mainly show strong acidic sites", but in fact, the modification of Fe can also lead to changes in the functional groups of biochar, please refer to Cui B, Chen Z, Wang F, et al. Facile Synthesis of Magnetic Biochar Derived from Burley Tobacco Stems towards Enhanced Cr (VI) Removal: Performance and Mechanism[J]. Nanomaterials, 2022, 12(4): 678.

Author Response

Response to Reviewer 1 Comments

1.line 291, "The modification of Fe can promote micro-mesoporous channels development, and mainly show strong acidic sites", but in fact, the modification of Fe can also lead to changes in the functional groups of biochar, please refer to Cui B, Chen Z, Wang F, et al. Facile Synthesis of Magnetic Biochar Derived from Burley Tobacco Stems towards Enhanced Cr (VI) Removal: Performance and Mechanism[J]. Nanomaterials, 2022, 12(4): 678.

First of all, thank you for your comments on our manuscript. We have referred to this article and made additions in the revised manuscript.

Reviewer 2 Report

The paper presented by Li et al. reviews the formation mechanism of biochar from lignocellulosic biomass pyrolysis. Also, the different activation methods are described.

Overall, the paper deals with an interesting topic for the scientific community. This reviewer thinks that the paper is well written and it has enough quality to be published after some minor modifications.

This paper could be improved with the incorporation of more recent works in the field. Some of the references used are very old. Please, carefully revise this. 

Please, provide a typos and grammatical errors free version of your work (e.g., L 186 or L 245 among many others).

Author Response

Response to Reviewer 2 Comments

1.This paper could be improved with the incorporation of more recent works in the field. Some of the references used are very old. Please, carefully revise this. 

First of all, thank you for your comments on our manuscript. We have cited more recent literature and have made additions in the revised manuscript.

2.Please, provide a typos and grammatical errors free version of your work (e.g., L 186 or L 245 among many others).

We spell-checked and grammar-checked the manuscript and made corrections, which can be viewed in the revised manuscript, thank you.

Reviewer 3 Report

In this manuscript, the authors reviewed that the formation mechanism of biochar from lignocellulosic biomass pyrolysis. Subsequently, the activation and modification methods of biochar catalysts, including physical activation, chemical activation, metal modification, and nonmetallic modification are summarized. Finally, the application of biochar-based catalysts for lignocellulosic biomass and waste plastics pyrolysis is discussed in detail and the catalytic mechanism of biochar based catalysts was also investigated. In overall, this manuscript is interesting but in order to consider publication, this work should be revised. The following comments should be addressed for the improvement of their manuscript.

Comment 1: The overall study aims for this review study about the value-added products from catalytic pyrolysis of lignocellulosic biomass and waste plastics over biochar-based catalyst need to be further clarified in detail as compared to other conventional formation systems for biochar.

Comment 2: What are the potential value-added products can be formed through the biochar-based catalysts for lignocellulosic biomass and waste plastics pyrolysis? Please provide a summary of recent literature on the potential value-added products can be formed through the biochar-based catalysts in term of

research findings and efficiency performance should be discussed.  Please discuss and clarify with fundamental support. 

Comment 3: The commercialization perspectives and marketing strategies for these potential value-added products from catalytic pyrolysis of lignocellulosic biomass and waste plastics over biochar-based catalyst can be further discussed in detail in the conclusion section.

Comment 4: The carefully English correction is necessary for the whole manuscript. Please check and revise accordingly.

Author Response

Response to Reviewer 3 Comments

Comment 1: The overall study aims for this review study about the value-added products from catalytic pyrolysis of lignocellulosic biomass and waste plastics over biochar-based catalyst need to be further clarified in detail as compared to other conventional formation systems for biochar.

Thank you for your suggestion, we have clarified this issue in the manuscript, please check this section in the revised manuscript.

Comment 2: What are the potential value-added products can be formed through the biochar-based catalysts for lignocellulosic biomass and waste plastics pyrolysis? Please provide a summary of recent literature on the potential value-added products can be formed through the biochar-based catalysts in term of research findings and efficiency performance should be discussed.  Please discuss and clarify with fundamental support. 

We have added this section in section 3.6 of the revision, please check it in the revision.  In addition, we have presented the composition of the value-added products of catalytic pyrolysis LB and WPs from BBCs in Table 3, Table 4, and Table 5 of the original manuscript. You can still see these in the revised manuscript now.

Comment 3: The commercialization perspectives and marketing strategies for these potential value-added products from catalytic pyrolysis of lignocellulosic biomass and waste plastics over biochar-based catalyst can be further discussed in detail in the conclusion section.

Thank you for your suggestion on the manuscript, which was added to the conclusion section of our manuscript, please check it in the revised manuscript.

Comment 4: The carefully English correction is necessary for the whole manuscript. Please check and revise accordingly.

We spell-checked and grammar-checked the manuscript and made corrections, which can be viewed in the revised manuscript, thank you.